# Changes in the Retail Food Environment in Mexican Cities and Their Association with Blood Pressure Outcomes

**DOI:** 10.3390/ijerph19031353

**Published:** 2022-01-26

**Authors:** Marina Armendariz, Carolina Pérez-Ferrer, Ana Basto-Abreu, Gina S. Lovasi, Usama Bilal, Tonatiuh Barrientos-Gutiérrez

**Affiliations:** 1Department of Public Health, University of Texas at San Antonio, San Antonio, TX 78249, USA; marina.armendariz@utsa.edu; 2National Council for Science and Technology (CONACYT), Mexico City 03940, Mexico; 3Instituto Nacional de Salud Pública, Cuernavaca 62100, Mexico; ana.basto@insp.mx (A.B.-A.); tbarrientos@insp.mx (T.B.-G.); 4Dornsife School of Public Health, Department of Epidemiology and Biostatistics and Urban Health Collaborative, Drexel University, Phildadelphia, PA 19104, USA; gsl45@drexel.edu (G.S.L.); ub45@drexel.edu (U.B.)

**Keywords:** food retail environment, blood pressure, Mexico, urban neighborhoods, nutrition policy

## Abstract

Shifting food environments in Latin America have potentially contributed to an increase in the consumption of ultra-processed foods and sugar-sweetened beverages, along with decreases in healthy foods, such as fruits and vegetables. Yet, little is known about the impact that such changes in the food environment have on blood pressure in low- and middle-income countries, including Mexico. We utilized individual-level systolic and diastolic blood pressure (SBP and DBP) measures from the 2016 Mexican Health and Nutrition Survey (ENSANUT, *n* = 2798 adults). Using an inventory of food stores based on the economic census for 2010 and 2016, we calculated the change in the density of fruit and vegetable stores, convenience stores, and supermarkets. Multilevel regression was used to estimate the association between the 2010–2016 food environment neighborhood-level changes with individual-level blood pressure measured in 2016. Declines in neighborhood-level density of fruit and vegetable stores were associated with higher individual SBP (2.67 mmHg, 95% CI: 0.1, 5.2) in unadjusted models, and marginally associated after controlling for individual-level and area-level covariates. Increases in the density of supermarkets were associated with higher blood pressure outcomes among adults with undiagnosed hypertension. Structural interventions targeting the retail food environment could potentially contribute to better nutrition-related health outcomes in Latin American cities.

## 1. Introduction

Unhealthy diets have been linked to many chronic health conditions, particularly cardiometabolic diseases such as obesity, diabetes, and cardiovascular disease (CVD) [1]. In recent decades, CVD mortality has declined in high-income countries, whereas CVD mortality has increased in low- and middle-income Latin American countries [2]. In Mexico, for example, CVD is the leading cause of death [3]. High blood pressure (hypertension) is one of the most important risk factors for CVD [3], and almost half a million new hypertension diagnoses occur among adults in Mexico every year [4]. Moreover, 40% of adults with hypertension remain undiagnosed, and 87% of diagnosed cases have uncontrolled blood pressure [4]. Increased prevalence of hypertension has been partially attributed to life-style factors, such as unhealthy diets, physical inactivity, heavy alcohol consumption, and tobacco use [2]. These behavioral factors, in turn, are shaped by the environments in which people live. As the built and food environments in low- and middle-income countries change as a result of globalization and economic development, research shows an increase in major diet-related health problems [2,5]. 

There has been a recent shift in the focus of interventions, from clinical and individual-level to structural interventions targeting nutrition and the food environment to address the increasing prevalence of diet-related non-communicable disease [6,7]. In 2014, the government of Mexico introduced a 10% tax to sugar-sweetened beverages and a tax to energy-dense foods to address obesity [8]. Then, in 2020, new warning labels for packaged foods and non-alcoholic beverages were introduced [9]. These policies have not been enough to halt the increasing obesity prevalence; therefore, identifying other modifiable structural determinants of diet and diet-related chronic diseases must be explored. The retail food environment may present an opportunity for intervention, but evidence on its association with diet and diet-related chronic diseases is still limited for Mexico and similar middle-income countries. 

### The Retail Food Environment

The potential influence of retail food environments on health outcomes and dietary behaviors has gained prominence in low- and middle-income countries, with attention to monitoring the availability or density of healthy and unhealthy food stores [10,11]. The retail food environment is characterized by the availability, accessibility, and affordability of healthy and unhealthy foods from food stores, which may drive healthy/unhealthy food purchases and consumption, thereby contributing to chronic disease risk [10]. Latin American settings have experienced shifts in the food environment, changing the availability of healthy versus unhealthy food options, due to an expansion of supermarkets, chain convenience stores, and fast-food chains [12,13,14]. Given the shifts in the food environment throughout Latin America, there is growing concern for understanding the role of these changes as potential population-level determinants of chronic disease and diet-related outcomes, beyond individual-level behaviors. 

The high and increasing burden from chronic diseases in Mexico has been partially explained by a recent rapid transition away from traditional meal preparation and consumption patterns [5]. Over the last two decades, the retail food environment in Mexico has experienced an influx of grocery supermarkets and chain convenience stores, and has begun to resemble that of high-income countries such as the United States [15]. For example, large supermarkets grew by 64% and convenience stores grew 142% between 2010 and 2018, with higher concentration of these stores located in urban areas of Mexico, suggesting a potential shift in dietary-related outcomes [16]. Consequently, the changing density of these food stores may contribute to greater availability of unhealthy products, such as ultra-processed foods, sugar-sweetened beverages, and sodium-packed snacks, which may increase chronic disease risk among Mexican populations [17]. For example, ultra-processed foods with high energy density and low nutritional quality have increased their relative importance in the diet, along with corresponding decreases in healthy foods, such as legumes, fruits, and vegetables [13,18]. Indeed, ultra-processed food consumption has been linked to unfavorable outcomes, such as metabolic syndrome and elevated blood pressure [19,20].

Studies in Latin America have focused on the link between the density of food store types and obesity or diet. Previous studies indicate there is a consistent association between a higher density of fruit/vegetable shops and favorable health outcomes and behaviors across LA countries [21,22,23]. For example, Duran and colleagues found higher fresh produce market density to be associated with more fruit and vegetable consumption in Brazil, yet the density of such healthier foods was not associated with less sugar sweetened beverage consumption [23]. Conversely, populations living in neighborhoods with a higher density of unhealthy food stores, such as convenience stores, have been observed to have poor obesity-related outcomes [24,25]. Further, a systematic review of food environments in Latin America found inconclusive evidence of an association between density of convenience stores or fast-food outlets with low quality nutrition or obesity [26].

In Mexico, a higher density of supermarkets was found to be associated with lower obesity risk, whereas other studies find no evidence of this association [22,26]. Despite the positive dietary implications of supermarkets for health, such as a variety of dietary choices and accessibility to more people, negative implications include encouraging the consumption of energy-dense or highly processed foods, especially among marginalized populations [12], suggesting that findings may vary across studies. 

Recent work in Mexico has found that people living in neighborhoods where both density increases of fruit and vegetable stores decreased and chain convenience stores increased over a 6-year period had higher odds of diabetes; these findings were not observed in relation to changes in supermarkets [27]. Moreover, recent evidence from Pineda and colleagues showed a positive association between the density of convenience stores and BMI among a nationally representative sample of Mexican adults [28]. The implications of previous findings suggest that the high density of these food store options raise concerns for an increased sodium intake among Mexican populations, where the main dietary sources of sodium are foods found in convenience stores and supermarkets, including breads, meats, pizzas, sandwiches, cheese, and other packaged foods [29]. However, the mixed evidence regarding convenience stores and supermarkets, especially in contrast with the consistent findings pertaining to fruit and vegetable stores, warrants further investigation within the context of changing food environments.

Despite the growing literature on food environment research in Mexico, there is limited knowledge on the health impacts of changes in the food environment for Mexican populations, particularly for outcomes beyond obesity and diabetes [26,27]. In the present study, we seek to expand on previous LA findings, particularly in Mexico, where both changes in the retail food environment and blood pressure are underexplored. Although blood pressure is complex and dynamic, it is a worthwhile health measure to assess, given its strong links to CVD-related outcomes [30]. 

A cross-sectional approach was employed to examine the association between neighborhood-level density changes in food store types from 2010 to 2016, and blood pressure in 2016 in Mexico, and to identify potential heterogeneity in this association by hypertension status. Thus, we also examine the potential effect modification of hypertension status (described later as: (1) non-hypertensive, (2) undiagnosed hypertensive, and (3) diagnosed hypertensive) to distinguish between groups that may be more vulnerable to such changes in the food environment. 

## 2. Materials and Methods

### 2.1. Setting

This study was conducted as part of the *Salud Urbana en América Latina* (SALURBAL) study (see original study for design and methods) [31,32]. Cities have been operationalized by SALURBAL as agglomerations of administrative units that share a common urban built-up extent, with a population of 100,000 or more. Using SALURBAL’s operationalization of a city, we restricted our analysis to the 53 cities of Mexico with individual-level health data. This included residents of 147 neighborhoods, defined as *Áreas Geoestadísticas Básicas* (AGEB, similar to US census tracts). An urban AGEB is made up of blocks (1 to 50), and delimited by streets or avenues [33].

### 2.2. Data Sources

#### 2.2.1. Individual-Level Data

Individual-level survey data were obtained from the 2016 *Encuesta Nacional de Salud y Nutrición* (ENSANUT, translated to English as National Health and Nutrition Survey), a cross-sectional population-based household survey carried out to collect information on nutrition, health, and health-related services and interventions. The design of the sample included stratification and multistage sampling to ensure representativeness at the national, regional, and urban/rural levels [34]. Health, socioeconomic, and demographic modules were completed by 8294 adult men and non-pregnant women aged ≥ 20 years in 2016. 

The sample was restricted to adults living in the SALURBAL-defined cities (*n* = 3067). Non-urban areas were excluded due to concerns that the commercial food establishments in the food environment database would not be as comprehensive in capturing the relevant data, as rural food environments have a higher density of food cultivation and informal food establishments. Observations with incomplete information on the variables of interest were removed (*n* = 269), resulting in a final analytic sample of 2798 adults aged 20 years-old and older living in 147 AGEBs. 

#### 2.2.2. Retail Food Environment Data

Retail food environment data from 2010 and 2016 were obtained from the *Directorio Estadístico Nacional de Unidades Económicas* (DENUE, translated to English as National Statistical Directory of Economic Units), which is an inventory of five million non-itinerant economic establishments related to manufacturing, commerce, and services, characterizing their main economic activity and location [35]. The information for this directory is based on the National Economic Censuses, which is collected every five years (2009–2019) and updated annually using surveys and field work conducted by Mexico’s National Institute of Geography and Statistics (INEGI) [35,36]. Stores are georeferenced by Census field workers. Codes for state, municipality, and AGEB are included for every economic unit in publicly available databases.

Food store types were classified using the North American Industrial Classification System (NAICS) [37] as follows: (1) fruit/vegetable stores—defined as semi-permanent establishments that exclusively sell fruits and vegetables; (2) convenience stores—defined as open ≥ 18 h a day for 365 days a year, and sell mainly processed and ultra-processed food products and beverages (e.g., OXXO, 7-Eleven); and (3) large supermarkets—defined as chain grocery stores that sell both healthy and unhealthy food options (e.g., Superama, Bodega Aurrera). Chain convenience stores were searched by name, since NAICS codes do not allow for specific identification of this store type. Six companies that control 90% of the market were included in the chain convenience store classification: OXXO, 7-Eleven, Extra, Circle K, Bodega Aurrera Express, and Chedraui Supercito [38]. See Appendix A Table A1 for details of the NAICS food environment classification codes. 

### 2.3. Measures

#### 2.3.1. Blood Pressure

The primary outcomes of interest were systolic and diastolic blood pressure (SBP and DBP, respectively). In the ENSANUT 2016, two SBP and DBP measures were recorded and then averaged. Additionally, we assessed hypertension status as a potential effect modifier in the association between change in food store type and continuous blood pressure measurement. A respondent was considered to have hypertension if: (1) they self-reported *yes* to “Have you ever been told by a medical doctor that you have high blood pressure?”, or (2) had elevated measured SBP or DBP (using thresholds of =>140 mmHg SBP and =>90 mmHg DBP) [39]. To assess the effect modification by hypertension status, the following three categories were created: (1) non-hypertensive, (2) undiagnosed hypertensive, and (3) diagnosed hypertensive. 

Further, hypertension treatments and adherence may effectively modify blood pressure levels, potentially masking the effects of the food retail environment. Thus, we present supplementary findings where we stratify the association according to treatment among diagnosed adults with hypertension. Treatment status among individuals with hypertension was based on responses to the following question in ENSANUT 2016: “Are you currently taking medication (pills) to control your blood pressure?”, then dichotomized in the present study. A new category variable was created for our sensitivity results: (1) non-hypertensive, (2) undiagnosed hypertensive, (3) untreated hypertensive, and (4) treated hypertensive. 

#### 2.3.2. Food Store Density Changes 

We examined density changes in the food environment at the neighborhood-level as the exposure variable in 147 AGEBs (defined as neighborhoods in the present study). To estimate this, we first aggregate the number of stores by type at the AGEB level. Then, the density of each store type was operationalized by dividing the number of establishments in a neighborhood (AGEB) by the land area in km^2^. The density for each store type was calculated for 2010 and 2016. Based on the difference in density from 2010 to 2016 (6-year difference), we classified food store density changes in each neighborhood into one of the following change categories: (1) stable (no change), (2) decrease in density, or (3) increase in density. In the case of supermarkets and chain convenience stores, we collapsed stable and decline into a single category (no increase), since the number of neighborhoods with declining stores was minimal [27].

#### 2.3.3. Individual-Level Covariates

We used data on sex (male or female), age (in years, continuous), a household wealth index constructed from household assets and characteristics using a principal components analysis (categorized in tertiles), and educational attainment (dichotomized as incomplete high school, completed high school, or above). The wealth index was constructed using a principal components analysis from household assets and characteristics, e.g., ownership of refrigerator, vehicle, computer, pay TV, internet connection, washing machine, household floor material, number of rooms, number of lightbulbs, and water source. Household wealth indexes are commonly used in low- and middle-income countries as a proxy for consumption expenditure [39].

#### 2.3.4. Area-Level Covariates

Area-based population composition data were linked to individual records using a unique AGEB identifier. We selected contextual covariates that may be related to both changes in the density of food stores and blood pressure. We used data from the 2010 Census and the State and Municipal Database System (INEGI, 2010) to derive a marginalization index and the population density (population/km^2^) for each AGEB unit [40]. The marginalization index is a standardized composite index of ten socioeconomic variables at the AGEB level within four domains: education, health, household, and ownership of assets. The index has a mean of zero, and standard deviation of one. Negative numbers indicate lower marginalization, whereas positive numbers indicate a higher marginalization level. The index was categorized into three levels (high and very high, medium and low, and very low). 

### 2.4. Analytic Strategy

The primary objective was to examine the association between longitudinal changes in fruit and vegetable store density, changes in supermarket density, and changes in chain convenience store density at the neighborhood-level in relation to SBP and DBP among three groups (non-hypertensive, undiagnosed hypertensive, and diagnosed hypertensive). For descriptive purposes, we first displayed sample characteristics for the total sample, and for strata based on hypertension status. Next, we used multilevel linear regression models with a random intercept for neighborhood to examine the association between changes in the density of food stores (2010–2016) and SBP and DBP, estimated separately.
$$y_{ij}=\beta_{0}+\sum_{k=1}^{n}\beta_{k}X_{ki}+\sum_{l=1}^{m}\beta_{l}Z_{lj}+\varepsilon_{ij}+u_{j}\;$$
where ***y_i_*_j_** is the outcome (SBP or DBP) for participant ***i*** in neighborhood ***j***. ***X_i_*** is the set of explanatory variables at the individual level (level 1), and ***Z_j_*** is the set of explanatory variables defined for the neighborhoods (level 2). ***u_j_*** is the random intercept for level 2, and $\varepsilon_{ij}$ is the random error for the outcome, for which it is assumed that they are independent and follow a normal distribution with mean 0 and variance $\sigma_{u}^{2}$.

Each model was adjusted progressively to account for individual-level characteristics and area-level factors, including the 2010 density of each store type (the baseline level for change). We also explored the effect modification by hypertension status, in which interaction terms were added to the fully adjusted models. 

In sensitivity analyses, we examined whether stratification by treatment status (i.e., diagnosed and taking medication) may affect the association between the food environment and blood pressure outcomes. For example, even among those taking blood pressure medication, 55% are not achieving recommended blood pressure control, suggesting uncontrolled blood pressure may occur even among those taking medication [3]. Therefore, treatment status was examined as a potential effect modification variable. 

All analyses were conducted in STATA 16 (StataCorp, College Station, TX, USA) [41]. Descriptive indicators account for the complex survey design and survey weights. In statistical models, we do not use survey weights due to evidence indicating the complexity of using them when fitting multilevel models, and minimal differences from scaled weighted and unweighted estimates [42].

## 3. Results

### 3.1. Descriptive Statistics

Table 1 shows the characteristics of the included ENSANUT 2016 participants (*n* = 2798). Mean SBP for the total sample were 120.8 mmHg, and mean DBP were 73.8 mmHg. Moreover, 26.5% of the total sample had hypertension: 16.7% were diagnosed, and 9.8% undiagnosed. Among diagnosed participants, 77% were taking BP medication (not shown). The mean age was 42.1 years old, a majority of the sample had less than a high school education (66.1%), and 13.2% of the sample belonged to the poorest wealth category. With respect to the neighborhood level characteristics (N = 147 unique AGEBs), the mean population density was 10,469 persons per km^2^, and 46 (30.9%) AGEBs had very low or low marginalization level, 52 (34.9%) had medium, and 51 (34.2%) had high or very high marginalization. AGEBs in this study had on average 19 individual observations each (min 1, max 40). 

Table 2 displays the descriptive characteristics of neighborhood food environment changes (2010 to 2016) and corresponding individual-level SBP and DBP values in 2016. From 147 AGEBs included in the study, 39.5% experienced an increase, and 25.2% experienced a decrease in fruit and vegetable stores from 2010 to 2016, whereas 35.4% saw no change in density. For neighborhoods with decreases in fruit and vegetable shops, respondents’ values of SBP and DBP were higher, on average, compared to neighborhoods with no change. For convenience stores and supermarkets, decreases were very rare (not shown). Nevertheless, in 19.1% of AGEBs, there was an increase in the density of convenience stores, and in 4.2% of AGEBs, there was an increase in the density of supermarkets. Both SBP and DBP values were similar among people living in AGEBs with increases versus those without change in density of convenience stores/supermarkets. 

### 3.2. Multilevel Linear Regression

Table 3 and Table 4 show the associations between changes in food store type and both SBP and DBP measures, respectively. In Table 3, Models 1 and 2 suggest an association between density declines in fruit and vegetable stores and SBP. SBP was, on average, 2.67 mmHg higher (95% CI 0.14, 5.19) in neighborhoods where fruit/vegetable store density declined compared to SBP in neighborhoods with no fruit/vegetable density increase. Upon adjusting for person-level characteristics in Model 2, the association between density declines in fruit and vegetable stores and SBP remained significant (1.92 mmHg, 95% CI 0.06, 3.78); the association was attenuated once we adjusted for neighborhood-level variables. We did not observe an association between convenience stores or large supermarket density changes with SBP. As shown in Table 4, DBP was not associated with changes in fruit and vegetable shops, convenience stores, nor supermarkets.

Lastly, we examined whether the association between 6-year changes in the food environment and blood pressure varied by hypertension group. Figure 1 indicates that the association between density increases in large supermarkets and higher SBP was stronger among the undiagnosed hypertensive group (19.3, 95% CI 10.3, 28.4, p for interaction < 0.001) compared to the non-hypertensive group. However, the association between both fruit/vegetable shops and convenience store density changes and blood pressure did not vary by hypertension status. Additionally, the associations between 6-year change and DBP by treatment groups were similar (data not shown). Among the three treatment status groups, neighborhood density increases of large supermarkets were associated with increased blood pressure outcomes only among adults with undiagnosed hypertension.

## 4. Discussion

The present study examined neighborhood changes in three types of food retail stores from 2010 to 2016 in Mexico, and their association with arterial blood pressure measured in 2016. We also distinguished between non-hypertensive, undiagnosed hypertensive, and diagnosed hypertensive, to consider potential effect modification. The following key findings were identified: (1) density declines in fruit and vegetable shops are associated with higher SBP, though the association was attenuated after controlling for area-level covariates; (2) in adjusted models, neighborhood density increases in large supermarkets was associated with higher blood pressure among adults with undiagnosed hypertension. 

Food environment studies in Latin America have consistently found that a higher density of fruit and vegetable stores are associated with positive health outcomes, such as a lower prevalence of obesity and diabetes, as well as favorable dietary behaviors [12,15]. Our primary findings indicate that density declines in fruit and vegetable shops are marginally associated with higher blood pressure. Greater attention to the declining presence of healthy food stores, such as fruit/vegetable stores, is necessary, especially if it occurs in conjunction with increases in food stores that sell unhealthy foods, such as sodium-packed snacks and/or ultra-processed foods. Indeed, a previous investigation found neighborhoods in Mexico that simultaneously experienced a decline in fruit and vegetable stores and an increase in convenience stores were associated with diabetes among residents [29]. Urban policies should consider the unfavorable health effects associated with removing accessible fruit and vegetable establishments, and prioritize maintaining them, since fruit and vegetable stores have consistently played a protective role in Latin America [29]. 

Contrary to expectations, we did not observe an independent association between density changes in convenience stores and blood pressure. Although convenience stores represent a small proportion of the total number of stores (1.3% in 2016), it is the fastest growing store type in Mexico [43]. Despite our null findings, there is recent evidence to suggest a higher density of convenience stores may be a better predictor of other diet-related outcomes, such as higher body mass index, among both Mexican adults and children [27,44]. Thus, continued monitoring of their growth and impact are needed, especially given the associations with unfavorable health outcomes reported in previous studies. 

Next, we observed a null independent association between supermarket density changes and blood pressure; however, there was evidence of an effect modification by hypertension status. That is, we observed an association between increases in supermarket density and higher blood pressure only among those with undiagnosed hypertension. Although previous work in Mexico found an inverse association between higher supermarket density and a reduction in body mass index [22], the present null findings between supermarket density changes and blood pressure warrant continued investigation. Large supermarkets sell high-energy food and sugar-sweetened beverages, alongside fruits and vegetables, calling into question the hypothesis that supermarket availability may reduce chronic disease risk. 

We speculate that the heterogeneity of food selection and/or availability within large supermarkets may facilitate unhealthy food purchases and consumption among Mexican adults [7,29]. In turn, those with undiagnosed hypertension may be more susceptible to an increase in the supply of unhealthy food (i.e., associated with a new supermarket), and the selection of ultra-processed, sodium-packed, and or high-density foods that contribute to elevated blood pressure. Conversely, we speculate those who are aware of their hypertension status may make different dietary choices that are in line with their health needs. Unlike natural experiments conducted in the United States, our study cannot offer policy-level recommendations regarding the opening and closing of supermarkets [45]. Future research may benefit from using natural experiments to better understand not only changes in the food environment, but also the causal pathways that impact nutrition-related outcomes and the choice of food shopping venues [45,46]. 

### 4.1. Sensitivity Analysis

Furthermore, pharmacological blood pressure treatment is highly effective in controlling blood pressure when taken as prescribed [47]; however, uncontrolled blood pressure occurs even among those taking medication [4,30]. Thus, we considered whether further stratification by treatment status affects the association between the food environment changes and blood pressure. Still, treatment status did not change the main results. Changes in supermarket increases were associated with higher blood pressure among the undiagnosed group, consistent with the main analyses. 

### 4.2. Strengths and Limitations

We highlight the utilization of arterial blood pressure as a principal objective measure of chronic health risk, rather than solely hypertension prevalence. Hypertension coverage of diagnosed cases is poor in Mexico, and may be subject to respondent recall bias [4]. This may lead to the potential exposure misclassification, and contribute to null associations in population studies. For example, in a preliminary analysis, we did not observe a significant association between food environment changes and hypertension prevalence in 2016 (Appendix B Table A2). In turn, blood pressure may demonstrate greater sensitivity in the assessment and tracking of disease progression, as inadequate blood pressure control and lack of diagnoses are concerns, given increasing hypertension prevalence in Mexico and other middle-income contexts [3,4]. 

This study is not without limitations. First, our outcome assessment is cross-sectional; thus, temporality in the associations cannot be established. Future studies using longitudinal data or natural experiments are needed to examine the effects of the Latin American retail food environment on nutrition health outcomes to build on existing findings. Second, we have food retail environment data between 2010 and 2016, missing part of the food environment transformation that occurred in Mexico between the 1990s and 2000s, presenting limitations in our ability to report the impact of growth over a longer period [15]. Additionally, we were unable to measure changes in the availability of foods within stores, which may point to potential measurement errors, since there could be significant heterogeneity in healthy versus unhealthy options within these types of stores [48]. Further limitations point to the utilization of census tract equivalents to assess area-level exposure, and our study is limited to Mexican populations living in cities with more than 100,000 inhabitants. This level of exposure also meant limited variability in change over the period for a large number of neighborhoods. Nevertheless, the census tract is a more precise measure of exposure compared to aggregating information at the municipality-level or higher. 

For the purpose of our study, we did not include examinations of individual-level characteristics, such as the consumption of sodium, caloric intake, or the effects of stressful circumstances as contributing factors to our findings. For example, individual dietary intake (i.e., sodium intake) may mediate observed associations. Additionally, food environments such as local street vendors and local fast-food outlets may contribute to the number of calories consumed and/or the dietary intake of ultra-processed foods [49], and potentially influence changes in blood pressure. However, we were not able to include these investigations because informal food stalls are not part of DENUE. Future work in Mexico could focus on better characterizing the informal food environment to assess how it contributes to dietary behaviors in the Mexican population. Furthermore, residents’ interactions with their food environment are complex and can be influenced by several factors beyond their local context, including work schedules, time constraints, food prices, personal mobility, and safety—to name a few [50,51]. Given the link between persistent stress and overall cardiovascular risk [52], we point to psychosocial measures as additional considerations in future population studies to improve our understanding of individual changes in blood pressure.

## 5. Conclusions

In conclusion, we point to the changing food environment as a potential population-level determinant of health, particularly for individuals living in urban neighborhoods. Our findings showed a marginal effect in declines in fruit and vegetable stores and higher blood pressure in Mexican cities. We also found neighborhood density increases in large supermarkets were associated with higher blood pressure among adults without a hypertension diagnosis. To our knowledge, this is one of the first studies in Mexico to examine this association, adding to the emergent work that has already begun to examine changes in the retail food environment and other common nutrition-related diseases. Future studies should apply longitudinal study designs to help identify both direct and indirect causal pathways linked to nutrition-related diseases. Although the present findings are not generalizable beyond these Mexican cities, our findings could be expanded on and confirmed using different study designs in Mexico and Latin America. Structural interventions targeting the retail food environment could potentially contribute to better nutrition-related health outcomes in Latin American cities.

## Figures and Tables

**Figure 1 ijerph-19-01353-f001:**
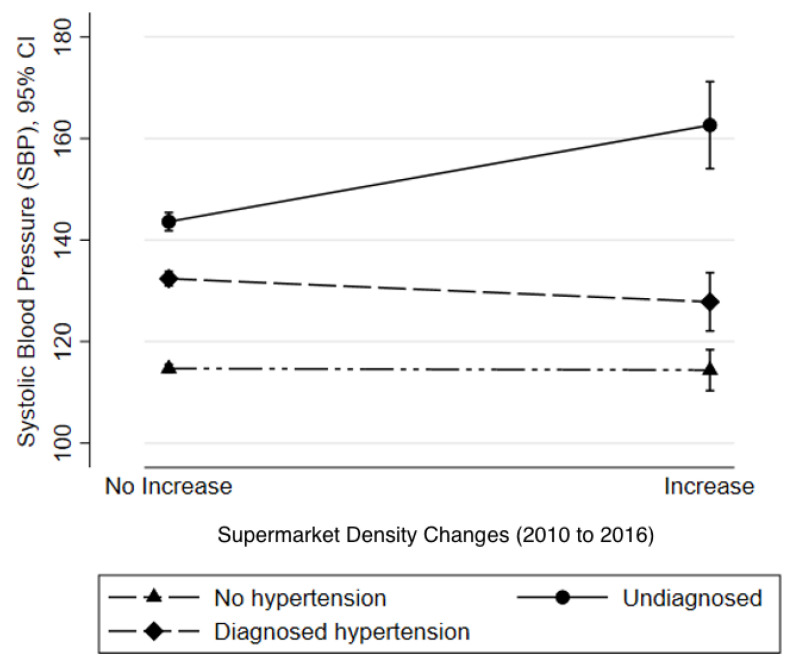
Predicted SBP means from the adjusted model. The association between 6-year supermarket density changes (no increase and increase) and blood pressure varied by hypertension group status (interaction *p* < 0.001).

**Table 1 ijerph-19-01353-t001:** Descriptive characteristics of adults with arterial blood pressure measurements (N = 2798).

	Hypertension Status			
	Total Sample	Non- Hypertensive	Undiagnosed Hypertensive	Diagnosed Hypertensive
	N = 2798	(1959)	(297)	(542)
Person level	Mean (SE) or % (SE)			
Blood Pressure				
SBP	120.8 (0.7)	113.5 (0.5)	146.7 (1.4)	137.4 (2.4)
DBP	73.8 (0.4)	70.5 (0.4)	87.9 (0.9)	80.1 (0.9)
Age	42.1 (0.6)	38.0 (0.5)	51.8 (1.3)	54.0 (1.5)
Female, %	50.7 (1.6)	49.7 (1.6)	39.5 (3.4)	61.9 (5.5)
Education, %				
Incomplete H.S. or less	66.1 (1.9)	61.8 (2.2)	77.9 (3.0)	78.0 (5.6)
Wealth Index, %				
Poorest (Tertile)	13.2 (1.3)	13.2 (1.4)	18.4 (4.1)	10.4 (2.2)
Middle (Tertile)	27.2 (2.2)	25.8 (2.3)	33.3 (5.7)	30.0 (4.8)
Richest (Tertile)	59.6 (2.4)	61.1 (2.7)	48.3 (5.1)	59.5 (5.2)

Abbreviations: SBP = Systolic blood pressure; DBP = Diastolic blood pressure.

**Table 2 ijerph-19-01353-t002:** Descriptive characteristics of neighborhood (AGEB, *n* = 147) food store density * changes (2010 to 2016) and corresponding individual-level (*n* = 2798) systolic blood pressure (SBP) and diastolic blood pressure (DBP) outcomes in 2016.

	AGEB		Blood Pressure Outcomes		
				SBP	DBP
Density Changes	N	% Change	N ^♦^	Mean (SE)	
Fruit/vegetable					
Decrease	37	25.2	651	122.5 (1.5)	74.8 (0.9)
No increase	52	35.4	1030	119.5 (1.3)	72.8 (0.7)
Increase	58	39.5	1117	120.9 (1.0)	74.3 (0.4)
Convenience stores					
No increase	119	80.9	2251	120.3 (0.8)	73.5 (0.4)
Increase	28	19.1	547	122.7 (1.5)	75.2 (0.9)
Supermarkets					
No increase	141	95.9	2680	121.0 (0.7)	74.0 (0.4)
Increase	6	4.1	118	117.3 (4.9)	71.0 (2.2)

^♦^ Individual level N. * Food store density = number of food stores by type/area in km^2^. Abbreviations: AGEB = Áreas Geoestadísticas Básicas (the equivalent of a US census tract).

**Table 3 ijerph-19-01353-t003:** Multilevel linear regression models of neighborhood-level density * changes of food store type and systolic blood pressure (SBP) measures for total sample (*n* = 2798).

Density Change	Model 1	Model 2	Model 3
Fruit and vegetable shops			
(1) Decrease	2.67 (0.14, 5.19)	1.92 (0.06, 3.78)	1.76 (−0.15, 3.67)
(2) No increase (ref.)			
(3) Increase	0.77 (−1.43, 2.98)	0.75 (−0.87, 2.52)	0.64 (−0.98, 2.61)
Convenience stores			
(1) No increase			
(2) Increase	0.71 (−1.72, 3.14)	0.72 (−1.06, 2.51)	0.82 (−0.94, 2.60)
Supermarkets (large)			
(1) No increase			
(2) Increase	0.77 (−4.06, 3.14)	0.99 (−3.08, 5.06)	0.53 (−3.03, 4.10)

Note: Model 1 is an unadjusted random intercept model; Model 2 is further adjusted by person-level age, sex, education, wealth index, and hypertension status; Model 3 is further adjusted by neighborhood-level population density, marginalization index, and 2010 density of fruit and vegetable stores, convenience stores, and supermarkets. Results shown are β coefficients (95% CI) representing the change in SBP (in mmHg) compared to areas with no change (reference). * Food store density = number of food stores by type/area in km^2^.

**Table 4 ijerph-19-01353-t004:** Multilevel linear regression models of neighborhood-level density changes of food store type and diastolic blood pressure (DBP) measures for total sample (*n* = 2798).

Density Change	Model 1	Model 2	Model 3
Fruit and vegetable shops			
(1) Decline	1.19 (−0.01, 2.38)	0.93 (−0.08, 1.94)	0.82 (−0.24, 1.88)
(2) No increase (ref.)			
(3) Increase	0.64 (−0.39, 1.69)	0.52 (−0.35, 1.41)	0.48 (−0.42, 1.37)
Convenience stores			
(1) No increase			
(2) Increase	0.45 (−0.70, 1.59)	0.31 (−0.64, 1.28)	0.37 (−0.60, 1.35)
Supermarkets (large)			
(1) No increase			
(2) Increase	−0.93 (−3.21, 1.35)	−1.70 (−3.63, 0.23)	−1.46 (−3.42, 0.51)

Note: Model 1 is an unadjusted random intercept model; Model 2 is a mixed-effects adjusted model for person-level age, sex, education, wealth index, and hypertension status; Model 3 is fully adjusted for the covariates in Model 2 and neighborhood-level population density, marginalization index, and 2010 density of fruit and vegetable stores, convenience stores, and supermarkets. Results shown are β coefficients (95% CI) representing the change in SBP (in mmHg) compared to areas with no change (reference).

## Data Availability

The SALURBAL project welcomes queries from anyone interested in learning more about its dataset and potential access to data. To learn more about SALURBAL’s dataset, visit https://drexel.edu/lac/ or contact the project at salurbal@drexel.edu.

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
