# Peer review of "Changes in the Retail Food Environment in Mexican Cities and Their Association with Blood Pressure Outcomes"

_ijerph, 2022, doi:10.3390/ijerph19031353_

Round 1

Reviewer 1 Report

Thank you for your work on this project and responses to my comments and concerns.  I just have a few concerns related to the results.

  1. For Table 1, Wealth Index section is unclear.  The percentages should be totaling 100% across hypertension status or down the tertiles of wealth.  Please double check those numbers.
  2. For Table 2, N's for blood pressure measures should be added.
  3. For Table 4, there is no statistically significant findings and bolding should be removed as well as result section should remove the "marginally significant" terminology.  Model 2 of fruit and vegetable shops in Table 3 is marginally significant as lower bound is so close to zero, but no need to add "marginally".   As a side note, did you consider stratifying by wealth index for Tables 3 and 4?  Or at least tested for potential effect modification of wealth?
  4. For figure 1, the means should at the minimum be adjusted for age and sex.  Blood pressure and dietary shopping and eating habits are associated with age and sex.  You have major confounding in this figure.  Or you could age- and sex- standardize the means to the population of Mexico.  Also, what was the p-values for the interaction between hypertension status and change in food environment associated with blood pressure?
  5. For future study, you may consider investigating the dietary intakes instead of blood pressure associated with changes in the food environment.  Also is there no ENSANUT data for 2010?  Why not consider population level changes in blood pressure based on the changes in food environment?  That would have been interesting.
  6. Throughout the manuscript, associations described related to blood pressure is represented as "change in SBP" or "change in DBP" but this is misleading as there is only one blood pressure measure per person at 2016.  It would be better to edit to state associated with higher or greater SBP levels and so forth.

Author Response

Thank you for your work on this project and responses to my comments and concerns.  I just have a few concerns related to the results.

1. For Table 1, Wealth Index section is unclear.  The percentages should be totaling 100% across hypertension status or down the tertiles of wealth.  Please double check those numbers.

Response 1: thank you for noticing that error. A number in one cell in the Non-hypertensive column was wrong. We have now corrected it. All column percentages for wealth now add to 100%.

2. For Table 2, N's for blood pressure measures should be added.

Response 2: Thank you for this suggestion. We have now added N’s for blood pressure in table 2.

3. For Table 4, there is no statistically significant findings and bolding should be removed as well as result section should remove the "marginally significant" terminology.  Model 2 of fruit and vegetable shops in Table 3 is marginally significant as lower bound is so close to zero, but no need to add "marginally".   As a side note, did you consider stratifying by wealth index for Tables 3 and 4?  Or at least tested for potential effect modification of wealth?

Response 3: Thank you for noticing this, you are right. We have removed bolding in table 4, and have edited the paragraph that described these results. It now reads:

“As shown in Table 4, DBP was not associated with changes in fruit and vegetable shops, convenience stores nor supermarkets.”

4. For figure 1, the means should at the minimum be adjusted for age and sex.  Blood pressure and dietary shopping and eating habits are associated with age and sex.  You have major confounding in this figure.  Or you could age- and sex- standardize the means to the population of Mexico.  Also, what was the p-values for the interaction between hypertension status and change in food environment associated with blood pressure?

Response 4: Figure 1 shows predicted means from an adjusted model i.e. we use the margins command in Stata to predict and plot means from the model that adjusted for sex, age, education, wealth, population density, area-level marginalization and changes in fruit and vegetable stores and convenience stores. Perhaps the title of the figure was not explicit enough, we have now changed it to:

“Figure 1. Predicted SBP means from the adjusted model. The association between 6-year supermarket density changes (no increase and increase) and blood pressure varied by hypertension group status (interaction p<0.001)”

Note that we added the interaction p as per your comment above. We have also checked that the methods were correct and we believe they are. The section that refers to this analysis reads:

“Each model was adjusted progressively to account for individual-level characteristics and area-level factors including the 2010 density of each store type (the baseline level for change). We also explored effect modification by hypertension status, in which interaction terms were added to the fully adjusted models.”

5. For future study, you may consider investigating the dietary intakes instead of blood pressure associated with changes in the food environment.  Also is there no ENSANUT data for 2010?  Why not consider population level changes in blood pressure based on the changes in food environment?  That would have been interesting.

Response 5: Thank you for the suggestion. Unfortunately, there is no data on blood pressure for 2010 (ENSANUT 2010). In future work, we will consider dietary intake as well; however, we aimed to examine blood pressure because previous work has largely examined obesity and/or dietary outcomes.

6. Throughout the manuscript, associations described related to blood pressure is represented as "change in SBP" or "change in DBP" but this is misleading as there is only one blood pressure measure per person at 2016.  It would be better to edit to state associated with higher or greater SBP levels and so forth.

Response 6: Changes have been made in the abstract and throughout the results to reflect this suggestion. We have checked that when we say change we are only referring to the food environment (i.e. the exposure) and not BP.

Reviewer 2 Report

See attached file. 

Reviewer 3 Report

This article conducted an interesting survey aimed at revealing the association between Retail Food Environment and blood pressure outcomes. Large numbers of individual data and three statistical models were involved in the analysis. All the conclusions were draw based on data. However, although this article presented the Individual-level data in 2010, as well as Retail Food Environment Data from 2010 to 2016, it seems to lack an important data, namely the Individual-level data in 2010. The article tried to use the data in the groups of No increase as reference for comparison, but it is hard to ensure that the blood pressure in the groups of No increase did not significantly change from 2010 to 2016. What’s more, the blood pressure was affected by many other dominate factors, such as sleep, alcohol consumption and stress. Therefore, from our perspective, the blood pressure in the groups of No increase would change significantly, which is at least due to their increased ages. So please add the detail information in the groups of No increase such as the distribution of blood pressure, age, gender, education and wealth index, or the Individual-level data in 2010.

The other concern is that, the association between Retail Food Environment and blood pressure outcomes is not so significant. Although the density changes of fruit/vegetable significantly affected blood pressure in several models, increasing evidences show that fruit might be not so good for health due to the high contents of sugars such as fructose (Fructose stimulated de novo lipogenesis is promoted by inflammation. Nature Metabolism. DOI: https://doi.org/10.1038/s42255-020-0261-2). The conclusion of this article might come from the wrong basic knowledges. Consequently, it is better to classify the food in stores into more types, including meat, eggs, milk and snacks. At least, fruit and vegetable should be separated for further analysis.

Author Response

Reviewer 3:

1. This article conducted an interesting survey aimed at revealing the association between Retail Food Environment and blood pressure outcomes. Large numbers of individual data and three statistical models were involved in the analysis. All the conclusions were draw based on data. However, although this article presented the Individual-level data in 2010, as well as Retail Food Environment Data from 2010 to 2016, it seems to lack an important data, namely the Individual-level data in 2010. The article tried to use the data in the groups of No increase as reference for comparison, but it is hard to ensure that the blood pressure in the groups of No increase did not significantly change from 2010 to 2016. What’s more, the blood pressure was affected by many other dominate factors, such as sleep, alcohol consumption and stress. Therefore, from our perspective, the blood pressure in the groups of No increase would change significantly, which is at least due to their increased ages. So please add the detail information in the groups of No increase such as the distribution of blood pressure, age, gender, education and wealth index, or the Individual-level data in 2010.

Response 1: Thank you for the observations and feedback. We would like to begin by clarifying the following comment: “The article tried to use the data in the groups of No increase as reference for comparison, but it is hard to ensure that the blood pressure in the groups of No increase did not significantly change from 2010 to 2016”

  • It appears this comment suggests the “No Increase” group refers to BP; however, our “no increase” relates to changes in food stores, specifically FV stores, convenience stores, and supermarkets. This was assessed using objective data from DENUE 2010 to 2016. Therefore, we did not draw conclusions of BP overtime nor comment on BP trends because as you stated, we do not have individual level data from 2010 since it is not provided in ENSANUT. In turn, we applied a cross-sectional analysis utilizing the available data from ENSANUT 2016, where we observed BP was higher for the “Decrease” group for FV stores compared to the “No Increase” group for FV stores.
    • Please note that Table 2 presents the mean and SE for SBP and DBP stratified by density changes (i.e. decrease, increase, no change)

2. The other concern is that, the association between Retail Food Environment and blood pressure outcomes is not so significant. Although the density changes of fruit/vegetable significantly affected blood pressure in several models, increasing evidences show that fruit might be not so good for health due to the high contents of sugars such as fructose (Fructose stimulated de novo lipogenesis is promoted by inflammation. Nature Metabolism. DOI: https://doi.org/10.1038/s42255-020-0261-2). The conclusion of this article might come from the wrong basic knowledges. Consequently, it is better to classify the food in stores into more types, including meat, eggs, milk and snacks. At least, fruit and vegetable should be separated for further analysis.

Response 2:

The reviewer is right to point out that we found few statistically significant associations. This may be due to a number of potential reasons which we discuss. Not finding significant associations should not be of concern since, it is still worthwhile to document non-significant associations and avoid publication bias.

Regarding the comment about “fruit and vegetables not being so good for health”, we would like to point out that this is an epidemiological study and therefore, we draw from population-level research which consistently shows that vegetable and fruit consumption is protective of a number of chronic diseases and specifically of high blood pressure:

  • For example, systematic reviews and meta-analyses have concluded that diets containing fruit and vegetables are associated with lower SBP and DBP
    • Schwingshackl L, Schwedhelm C, Hoffmann G, Knüppel S, Iqbal K, Andriolo V, Bechthold A, Schlesinger S, Boeing H. Food Groups and Risk of Hypertension: A Systematic Review and Dose-Response Meta-Analysis of Prospective Studies. Adv Nutr. 2017, 8(6):793-803.
    • Ndanuko RN, Tapsell LC, Charlton KE, Neale EP, Batterham MJ. Dietary Patterns and Blood Pressure in Adults: A Systematic Review and Meta-Analysis of Randomized Controlled Trials. Adv Nutr. 2016, 7(1):76-89.
  • This epidemiological evidence has been taken up by dietary guidelines around the world which recommend consuming a high volume of FV, especially compared to processed foods. For example, the Mexican dietary guidelines recommend that one-third of dietary intake is comprised of FV, while EAT-Lancet suggests the consumption of FV as well as other plant-based options should double.
    • https://www.dof.gob.mx/nota_detalle.php?codigo=5285372&fecha=22/01/2013
    • https://eatforum.org/eat-lancet-commission/eat-lancet-commission-summary-report/

Based on this evidence we believe food environments must promote vegetable and fruit consumption and therefore our study is important.

We remind the reviewer that FV stores are indeed separate from other food store types in the analysis. In our methods, FV stores are defined as semi-permanent establishments that exclusively sell fruits and vegetables, which is a proxy for types of stores that sell unprocessed or healthier food options.

Round 2

Reviewer 1 Report

This is a great paper highlighting the importance of considering the context of places where people live in association with food environment and blood pressure.  It is okay to state that after adjusting for neighborhood level factors the association estimate between fruit and veggie shops and SBP was attenuated and lost significance, meaning that structural societal factors may be contributing to BP outcomes above and beyond the food environment.  Great work!  Wishing all the best in future studies.   

Author Response

Thank you for your encouraging remarks! 

Reviewer 3 Report

This article tried to reveal the association between Retail Food Environment and blood pressure outcomes. As a whole, the results were not sufficiently supported by the current data. This kind of study specially need adequate data and elaborate experiment design.

1) for adequate data: This article did not provide the detail information (at least blood pressure) in the groups of No increase in 2010, which might be unacceptable. A specific baseline for a biological biomarker is necessary and fundamental for a scientific study.

2) for elaborate experiment design: Although there are lots of reports showing the positive effects of fruit and vegetable on health, the authors should notice the increasing studies which revealed the side-effects of fruit. This article should include more kinds of food such as meat, eggs, milk and snacks into investigation. In addition, besides gender, age, education, wealth, a lot of other factors such as living habits (smoking, drinking, sleep, exercise), emotional status (stress, anxiety, depression), obesity and diabetes affected blood pressure. This article needs to think about more aspects. Interestingly, it seems that authors had known that fruit/vegetables must play a crucial effect on blood pressure, which falls into “arguing in circles”. Moreover, few statistically significant associations were found in this article. We also admit it is worthwhile to document non-significant associations and avoid publication bias, while it is based on a considerate experiment design. We think the experimental design is not so precise and thoughtful and thus the results are not convincing.

Author Response

Thank you for reviewing our study. To address concerns regarding methodology, we consulted with two statisticians regarding 1) the representativeness of both the exposure and outcome data and 2) the categorization of our outcome. Both statisticians confirmed our methods are robust to answer our study questions. Further, in our discussion as well as using a cross-sectional design with the secondary data that was available to ENSANUT.

Moreover, we previously addressed reviewers comments regarding the opposing view of fruit/vegetables being less beneficial. In our responses, we presented the following points:

  • Although we acknowledge fruit/vegetables contain sugar, we would like to address this suggestion by pointing the nature of this being a population-level study, not experimental; thus, we draw from population-level research given the nature of our sample and data collection.

This manuscript is a resubmission of an earlier submission. The following is a list of the peer review reports and author responses from that submission.

Round 1

Reviewer 1 Report

Interesting paper that aims to assess the associations between the changes in the retail food environment and blood pressure outcomes in Mexico. Despite the need for more evidence on this very important matter in LMIC, I am afraid the paper has a few limitations that need to be better addressed before it can be published. The reviewers have a few comments regarding the methods, the discussion, and the presentation of the results.
1.    Could the authors please provide more information about how they gathered the information to classified food retail outlets? What information was provided by the INEGI? Did they undertake in-store assessments or was ground-truthing undertaken (virtual or otherwise)? Did they use an existing tool to classify food outlets?

2.    Would be helpful to include some references to data/studies indicating the increasing density of fast-food chains, supermarkets, food retail density in general in Mexico.

3.    it might be useful to include some sort of definition of density in brackets ( i.e. count, count per 10,000 population, average per square kilometre) in tables.

4.    It is essential that the authors clearly state how density was calculated. This should be possible to replicate by others and is currently not possible to verify the results of this study.

5.    Have the authors adjusted the models for the sampling weights ENSANUT has? It is not clear. If not, this needs to be done.

6.    Please expand/add genetic susceptibility to the environment. Perhaps it may be more useful to discuss how the rapid change in blood pressure outcomes can’t be reflected by genetics alone and discuss the external forces at play. 

7.    Did the authors combine in the same group large and international food outlets (e.g.  hypermarkets, small grocery stores; fast-food chains with local fast-food outlets?) What were the advantages or disadvantages of such choice? In Brazil, for instance, despite the presence of international fast-food chains in most big cities, small locally owned fast-food outlets are still the most visited fast-food outlets in the country, in particular in low-income neighbourhoods where international chains are not as widely found. I would imagine Mexico to be similar. Please include this point in the discussion as a possible limitation of the study. 

8.    The introduction and discussion section lacks references to back up the author's statement throughout. For example, lines 48-50 require references to back up statements.  Line 82, says few studies, which few studies?

9.    Lines 94-95: Why was the study restricted to 53 cities – please explain and justify.

10.    Lines 114: 2,798 – this is not representative of the population, please adjust the title and manuscript to refer to these urban cities, rather than imply that it is the whole country that is being analysed. 

11.    Line 193. To avoid report selection bias, these results should be included in the appendix.

12.    How were the models designed? To justify explored models, please include mathematical equations of proposed methods in the methods section/statistical analysis. 

13.    Lines 156-163 require further details and clarity about the methods.

14.    Lines 334 – conclusion seems too bold as there is potential bias involved and a power deficit in the analyses. It must be highlighted that in-store food environments may have an impact on food selection. 

15.    Line 341 – the concept of ‘removing FV’ is mentioned widely. Perhaps a different framework would be beneficial where an increased FV offer that competes with ultra-processed foods in terms of availability/access and affordability is required. 

16.    Lines 343-345 are relevant as part of the discussion but not a conclusion of this study.

17.    Despite being not as numerous as those found for high-income countries, there are several studies on the food environment conducted in Mexico and other LMIC. Please revise the manuscript and include references. 

18.    Authors could strengthen their paper by discussing the limitations of their study, including the limitation presented by various papers that evaluated the impact of natural experiments related to the opening and closing of retail food stores in the US and elsewhere.

19.    The role of street markets, farmer's markets, as well as other types of eating venues (food trucks, street food, etc.) in the Mexican diet should be better specified/discussed if not included. If these venues play an important role in the number of calories consumed by Mexicans, the limitations of not including these venues in the analyses should be discussed. 

20.    How were the store addresses geocoded? Please make it clearer. Also, have the authors conduct any reliability analysis? 

21.    The choice to account for the census tract as the area of individual-level exposure can be troublesome and needs to be better justified by the authors and the limitations of such choice discussed. 

22.    Have the authors attempted to run models with standard deviations of store density? As the non-parametric distribution may lead to erroneous interpretations of the results.

23.    Have the authors attempted to correct the models by some sort of urbanicity or population density confounder? The depicted association might be capturing another unmeasured exposure to a very dense local urban environment. 

24.    Have the authors carried out sensitivity analyses?

Author Response

Thank you very much for your constructive comments and insightful suggestions for improving our work. Please see attached document with detailed comments. 

Reviewer 2 Report

This is a timely paper that is investigating neighborhood-level food environment changes related to blood pressure measures.  As epidemiologic research has begun to shift towards investigating more structural aspects of social determinants of health related to health outcomes, this project has the potential to contribute to the scientific evidence in this field.  However, there are some concerns and suggestions that, if addressed, would allow more robust investigation of these associations.

  1. If the study aim is to examine the association between changes in the food environment and blood pressure measures, it is not clear why people with undiagnosed hypertension would be separated from those with non-hypertension.  The group of diagnosed hypertension makes sense in that people in that group may be taking medication and/or eating differently based on their diagnosis that would influence their blood pressure.  However, separating the non-hypertension and undiagnosed hypertension groups just truncates the blood pressure distribution and reduces the power to detect "true" associations. 
  2. The findings related to the undiagnosed hypertension group's blood pressure positively associated with changes in supermarket density but not found in the diagnosed hypertension group (suggestions of negative association), could be just an artifact of hypertension awareness.  Those aware of their status may eat differently than those unaware, yet this is not mentioned in the discussion.
  3. More explanation is needed regarding the wealth index as to the variables used in the principal components analysis.  Also, more details on the marginalization index would be useful (the link for reference 23 is not an active link), especially whether a direction of the magnitude of marginalization.  For example, I assumed that a larger negative number means more marginalization based on table 1 but not sure how to interpret the marginalization index mean.
  4. To better address structural social determinants of health related to the food environment, it would be fruitful to examine the interaction between wealth index and food environment or the marginalization index and food environment.  Although I can appreciate the examination of education as an effect modifier, over-processed less healthy foods tend to be affordable and financial status will influence dietary decisions to a greater degree than education.  Therefore, having more convenience stores in these populations may be more detrimental than more affluent settings.
  5. Since there are no measures for blood pressure in 2010 to assess changes in blood pressure associated with changes in food environment, it may be useful to add the baseline food environment density value to the model to capture what the values were to begin with.  If the density was high to begin with, then changes do not capture the same amount of relative exposure.  This might be the case with supermarkets with such a small number of AGEBs showing an increase which most AGEBs may already have high densities.
  6. It would be nice to see mean food store density values for 2016 added to table 1.

This project is focusing on an important topic.  It just needs to be more fine tuned to consider some additional points mentioned.

Author Response

Thank you very much for your constructive comments and encouragement. Please see attached document with detailed comments. 

Reviewer 3 Report

This manuscript describes a retrospective cohort study, in which individual blood pressure information in 2016 and neighborhood food store change during 2010 to 2016 were analyzed to determine whether the food store change is one of the risk factors of hypertension in Mexico. This study has some novelty for the idea about food suppliers. But the retrospective cohort study requires the disease data on both 2010 and 2016 to measure the change on hypertension. In the following, I listed several major concerns need to be addressed.

  1. Could you give more evidence to illustrate the relationship between food store change and personal diet choice change?
  2. Could you prove the food store change is an independent variable, not disrupted by lifestyle or work stress? .
  3. In tables, full names of abbreviations should be showed in the footnote (such as AGEB, SBP).
  4. There is a typo in Figure 1 title, supermarket not supermarkey.

Author Response

Thank you very much for your constructive comments and insightful suggestions for improving our work. Please see attached document for detailed responses/comments. 

Reviewer 4 Report

Dear authors,

You have approached an interesting and important topic but I am concerned about the design and analyses and thus your interpretation of the results. Below are my main comments/questions.

  • to what extent are the neighborhood clusters also the clusters where the participants are living? It appears that the participants are representative of the cities n=53 and not of the clusters n=147. If this is correct the statistical modeling is incorrect and needs to be redone.
  •  
  • what is the justification to have “change” as the exposure and not look at variation between the different clusters? (Is it the change in food environment or the actual food environment which is associated with BP?) Thus, if exposure is based on a change, then the outcome should also be a change i.e a change in BP. Further, if the idea is “change” as an exposure it assumes that participants to large extent have been living in the clusters from 2010 to 2016 raising questions about mobility of the participants.  

  • comparison groups for “hypertension status” (non-BP, undiagnosed BP and diagnosed BP) does not make conceptually sense for the research question. In it’s construction it is mixing data on outcome measures (BP) with data on perceptions (self-reported diagnose), the latter which is a potential determinant/effect modifier. Further, how many of the self-reported “yes” actually had high BP? and which group where they categorized if they didn’t have high BP? For the apparent purpose of the variable it would be better to define “perceived BP status” as “yes” vs “no” as this is what could modify the behaviour.
  • Table 1
    • wealth tertiles does not appear to be tertiles from their distributions. How was these categorized? i.e basis for cut-offs?
    • Neighbourhood level characteristics should not be presented in strata of individual level categories suggesting an individual level sample size.
    • Neighbourhood characteristics should be based on cluster sample size n=147 and include number of participant per cluster (mean, sd, min, max), preferably also neigbourhood wealth (mean, sd, min max) which could be based on aggregated household data, and neighbourhood proportion of incomplete education (mean, sd, min, max)

  • table 2
    • are the BP measures representing the AGEB clusters? i.e is there a mean BP for each cluster which is then compiled to a mean for the categories? or?

  • table 3, 4
    • before I can interpret these tables I need to have an understanding of the representativeness of the BP data, i.e representing the cluster?
    • I suggest use of a new “perceived BP” variable instead of the one currently used
    • why is not the “increase” the reference for fruits? it is the “best”category.

  • Fig 2
    • the basis for the figure is unadjusted analyses and thus this needs to be taken into consideration in interpretation. i.e the results are potentially biased.

  • discussion:
    • the concerns/questions outlined above needs to be sorted out before comments on the discussion section can be made.

Author Response

(The authors gave the same response as above.)

Round 2

Reviewer 3 Report

This manuscript describes a cross-sectional study, in which individual blood pressure information in 2016 and neighborhood food store change during 2010 to 2016 were analyzed to determine whether the food store change was related to blood pressure status in Mexico. This study has some novelty for the idea about food suppliers and the authors made comprehensive explanations to my questions. In the following, I listed several major concerns.

  1. I hope the authors could declare the type of this study as cross-sectional study in the introduction part for the readers’ convenience.
  2. Modern food environment, like fast food stations and convenience stores, tends to appear in the areas with larger proportion of tertiary sector of the economy or service industry, where the residents are more likely to be white-collar workers. Such population tend to live in a high-pressure and fast-pace environment. This living environment determines their diet choice as fast food because they have less time to cook. At the same time this fast-pace lifestyle with high stress can affect their blood pressure. So which one is the real factor? I suggest the authors to avoid the possible compound bias that the blood pressure change is more due to the high-pressure fast-pace lifestyle, rather than the diet change coming with the fast-pace lifestyle.

Reviewer 4 Report

Dear authors, 

Thank you for your response to my questions, some of which are now clear. However, for some of the critical issues such as representativeness of exposure and outcome data and categorization of outcome I am not convinced that you are doing the correct things.